# Evaluation of Urinary miRNA in Renal Cell Carcinoma: A Systematic Review

**DOI:** 10.3390/cancers17081336

**Published:** 2025-04-16

**Authors:** Giovanni Cochetti, Liliana Guadagni, Alessio Paladini, Miriam Russo, Raffaele La Mura, Andrea Vitale, Eleonora Saqer, Paolo Mangione, Riccardo Esposito, Manfredi Gioè, Francesca Pastore, Lorenzo De Angelis, Federico Ricci, Matteo Mearini, Giacomo Vannuccini, Ettore Mearini

**Affiliations:** Urology Clinic, Department of Medicine and Surgery, Santa Maria della Misericordia Hospital, University of Perugia, 06129 Perugia, Italy; giovanni.cochetti@unipg.it (G.C.); liliana.guadagni@specializzandi.unipg.it (L.G.); miriam.russo@specializzandi.unipg.it (M.R.); raffaele.lamura@specializzandi.unipg.it (R.L.M.); eleonora.saqer@specializzandi.unipg.it (E.S.); paolo.mangione@specializzandi.unipg.it (P.M.); riccardo.esposito@specializzandi.unipg.it (R.E.); manfredi.gioe@specializzandi.unipg.it (M.G.); francesca.pastore@specializzandi.unipg.it (F.P.); lorenzo.deangelis@specializzandi.unipg.it (L.D.A.); federico.ricci@specializzandi.unipg.it (F.R.); matteo.mearini@specializzandi.unipg.it (M.M.); giacomo.vannuccini@unipg.it (G.V.); ettore.mearini@unipg.it (E.M.)

**Keywords:** microRNA, miRNA, renal cell carcinoma, RCC, biomarker, liquid biopsy

## Abstract

The role of microRNAs (miRNAs) as oncological biomarkers has been studied over the past ten years, with largely inconsistent results. In order to help researchers discuss the complexity of the subject and create standardized protocols for future study to combine findings from around the world, this systematic review attempts to collect data regarding the diagnostic power of urinary miRNAs in Renal Cell Carcinoma (RCC).

## 1. Introduction

Renal cell carcinoma (RCC) is the sixth most common tumor diagnosed in men and the tenth in women. It accounts for 3% of all cancers, with higher incidence in Western countries. In 2022, kidney cancer had an incidence rate of 4.4 new cases per 100,000 individuals worldwide, resulting in a total of 434,000 new diagnoses [1].

RCC ranks thirteenth as a cause of cancer-related mortality worldwide. In all types of RCC, prognosis worsens with stage and histopathological grade. One of the challenges in kidney cancer diagnosis is its silent behavior until advanced stages. Due to the asymptomatic course, especially in the early stages, about 60% of RCC are identified incidentally by abdominal US or CT [2].

Approximately 30% of patients are diagnosed with metastatic disease, underscoring the importance of early detection. Currently, the diagnosis of kidney cancer is predominantly based on radiological imaging such as ultrasound scan, computed tomography (CT) scans and magnetic resonance imaging (MRI). Also, renal mass biopsy can provide information, to characterize the mass and distinguish between benign and malignant lesions, thus guiding appropriate treatment decisions. In recent years, however, a new diagnostic modality has gained attention for its potential in kidney cancer: liquid biopsy. A minimally invasive test, liquid biopsy offers the ability to analyze biomarkers found in peripheral blood and urine, providing valuable insights into tumor characteristics without the need for tissue samples. This approach is particularly promising for diagnosing kidney tumors, determining their histological type, and assessing potential resistance to pharmacological treatments. Liquid biopsy encompasses several key biomarkers, including circulating tumor DNA (ctDNA), circulating tumor cells (CTCs), exosomes, and microRNAs (miRNAs) [3]. 

The latter are small, non-coding RNAs, typically 21–23 nucleotides long, which regulate gene expression by binding to the 3' untranslated region (3'UTR) of target messenger RNAs (mRNAs). This binding prevents the translation of the mRNA into protein, effectively inhibiting the expression of the gene. MiRNAs play a critical role in regulating key processes such as tumor initiation, invasion, metastasis, and resistance to therapy. Altered miRNA expression profiles have been observed in various cancers, including kidney cancer, making them potential biomarkers for early detection and prognosis. Importantly, circulating miRNAs have been shown to be remarkably stable in bodily fluids, such as blood and urine, making them ideal candidates for non-invasive diagnostic testing. Liquid biopsy, through the analysis of miRNAs and other biomarkers, holds great promise for improving the diagnosis and management of kidney cancer [4].

Liquid biopsy may become a cornerstone of kidney cancer diagnostics, offering significant advantages over traditional methods by enabling earlier, less invasive, and more personalized patient care. The aim of this systematic review was to investigate the most promising urinary miRNAs associated with RCC.

Our primary research question (RQ) is: What urinary miRNA are differentially expressed in adult patients with RCC compared to healthy individuals?

Our secondary RC is: What urinary miRNA expresses different behavior before and after surgery in RCC patients?

## 2. Material and Methods

The review was performed following the PRISMA (Preferred Reporting Items for Systematic reviews and Meta-Analyses) 2020 guidelines [5,6] integrated with the Synthesis Without Meta-analysis (SWiM) checklist [7] (Appendix A). The protocol of this systematic review was registered in PROSPERO (CRD42024550716) [8].

We established the following inclusion criteria adhering to the PICO framework (Population, Intervention, Comparator, Outcome).

-Population: Adult (≥18 years old) patients with RCC;-Intervention: Measurement of circulating or cell-free miRNA in urine samples of patients with RCC;-Comparator/Control: Healthy subjects or patients with RCC after surgery;-Outcome (main): Different expression of miRNA in urine samples between patients with RCC and healthy subjects through diagnostic accuracy measurements. Outcome (additional): Different expression of miRNA in urine samples of patients with RCC before and after surgery through diagnostic accuracy measurements.

The types of studies included were prospective cohort studies, randomized controlled trials, cross-sectional studies and case–control studies and case series.

The exclusion criteria were:-pediatric patients and adult patients with benign renal tumors;-measurement of RNA other than circulating or cell-free miRNA in blood samples of patients with RCC (snRNA, ccRNA, exosomal RNA, lncRNA…);-reviews and metanalysis, abstracts, letters and meeting report.

Our literature search strategies were deciding which database to include (Pubmed, EMBASE and Clinicaltrial.gov) and composing strings with words related to urinary miRNA in patients with RCC. 

The software Mendeley Desktop [9] was used to import bibliographic citations. Then, the results were exported in a Microsoft Excel spreadsheet used for title/abstract and full text selection [10]. 

The study selection process was performed by three independent review authors (LG, FP, PM). Initially the reviewers selected the records through the titles and abstracts according to the inclusion criteria. When disagreement incurred, it was solved through discussion and, when necessary, a supervisor (GC, AP) was involved. After that, four other review authors screened the full texts of the potential eligible studies. 

Eleven independent reviewers (LG, FP, MR, ES, AV, RLM, FR, PM, MG, LDA, RE) performed data extraction using a table including a list of extracted data. All the reviewers were thoroughly instructed on how to fill the table to secure consistency. Disagreements on data extracted were solved involving a supervisor (GC, AP).

We extracted from each study: -bibliographic data: first author, publication year and citation;-study characteristics: study design, country, number of centers, sample size;-participant characteristics: disease, gender, age;-intervention characteristics: type of miRNA, dosage method and phase;-control characteristics: healthy subjects or patients with RCC in other phases;-study outcomes.

We decided not to perform metanalysis because we expected excessive heterogeneity between studies in matter of type of miRNA, extraction method and normalizers used.

Whereas dichotomous data were aggregated as pooled risk ratio (RR), continuous data were aggregated as mean difference (MD). The inverse variance approach and the random effect model were adopted. We reported 95% confidence intervals. 

The risk of bias assessment was performed by two independent reviewers (LG, MR) that received clear instructions for the use of critical appraisal tools, Cochrane Risk of Bias Tool (RoB 2) for RCT and ROBINS-E (Risk Of Bias In Non-Randomized Studies–of Exposure) for non-randomized studies [11]. After conducting a pilot phase to ensure homogeneous evaluation, the records were classified low risk, some concerns, high risk or very high risk.

Certainty assessment was performed by a reviewer (LG) using GRADE (Grading of Recommendations Assessment, Development and Evaluation) method [12]. Since the papers were not comparable because they studied different miRNAs extracted and treated with different methods, we did not evaluate the overall inconsistency but discussed them singularly. We categorized the records in high, moderate, low or very low quality.

## 3. Results

The initial research strategy identified 593 studies (377 from Pubmed, 215 from EMBASE and 1 from Clinicaltrials.gov), 175 of them were duplicates and were excluded. Of the remaining 418 studies, 28 remained after title and abstract selection. On the basis on full text selection, 18 articles were excluded. The remaining 10 were included [13,14,15,16,17,18,19,20,21,22] in the final selection (Figure 1, PRISMA 2020 flow diagram). Appendix A lists the excluded studies and the reasons for their exclusion. The primary cause of exclusion was wrong outcome (11 records). Other reasons for exclusion were wrong study design, wrong population and wrong comparator.

We did not perform meta-analyses because the included studies were not uniform in terms of the kind of RCC (clear cell RCC, papillary RCC, not defined RCC), the normalizer used during the stabilization phase, and the miRNA that was investigated (showed in Figure 2). In Table 1, we outlined the main characteristics and findings of the included studies. Every record that was included was a diagnostic accuracy study.

A total of 10 studies, involving 611 patients, investigated expression levels of miRNAs and their diagnostic role in RCC. Following serum extraction, all studies normalized the miRNAs using either endogenous or synthetic controls. Cel-miR-39 was the most often used normalizer.

No miRNA was investigated in more than one paper by different authors, although miR-122, miR-1271-5p, miR-15b-5p and miR-210-3p were studied twice by the same authors [14,15,21,22].

One of the most consistently investigated miRNAs across the records was the **miR-210 family**, which showed significant upregulation in urine samples from RCC patients, particularly those with clear cell RCC (ccRCC) [17,21,22]. 

The **let-7 family miRNAs** (let-7a, let-7b, let-7c, let-7d, let-7d-5p, let-7e, and let-7g), were also frequently investigated [16,19]. The authors found that all let-7 miRNAs levels were significantly elevated in the urine of RCC patients compared to healthy controls. 

Another prominent miRNA was **miR-122**, which was found to be overexpressed in urine samples from ccRCC patients by Cochetti et al. [14,15]. Moreover, the authors developed the diagnostic algorithm 7p-urinary score that analyzed miR-122-5p, miR-1271-5p, and miR-15b-5p and three internal controls to better distinguish between miRNAs expression levels in cases and controls.

Other upregulated miRNAs were miR-15a, miR-30a-5p^me^, miR-34a-5p, miR-200a-3p, miR-205-5p, miR-365a-3p and miR-1275 [13,18,19,20].

Five papers reported different expression of miRNAs in urine samples before and after surgery: miR-15a, miR-34a-5p, miR-200a-3p, miR-205-5p, miR-210, miR-210-3p, miR-365a-3p and let-7d-5p levels decreased after nephrectomy, highlighting their possible utility in monitoring disease progression or response to treatment [17,18,19,21,22].

### Risk of Bias and Certainty Assessment for Included Studies

Since all of the included studies were non-randomized diagnostic accuracy studies, the ROBINS-E tool (Table 2) was used to evaluate the methodological quality [11]. 

Five low-risk studies and five records with some concerns of risk were included in the overall risk of bias during the critical appraisal phase. Most studies presented some concerns because no confounding factors were mentioned. 

The GRADE tool was used to complete the certainty assessment phase [12]. Non-randomized case–control studies were deemed good quality since all of the research used highly sophisticated laboratory equipment to detect miRNA urine levels objectively.

Studies that included at least 200 patients and controls in each phase were considered sufficient for numerosity because the variable considered were genetic materials. Only one study fulfilled this requirement, and, because of that, it was considered high quality [20]. Out of the remaining articles, five studies were deemed low quality, and four papers were categorized as moderate quality.

## 4. Discussion

This systematic review, encompassing 10 studies, provides a detailed examination of urinary miRNAs as potential diagnostic biomarkers for RCC, identifying the limitations of studies conducted to date and future prospects for improving the selection of these miRNAs to increase their diagnostic power. 

The miR-210 family was the most investigated as urinary biomarker. Both miR-210 and miR-210-3p were upregulated before surgery and down-regulated after surgery. This pattern of expression was noted in both tissue and urine samples, reinforcing the validity of the miR-210 family as a key player in RCC biology and its potential as a biomarker for early detection and follow-up. The miR-210 family is also the most investigated as serum biomarker in the literature [23].

The review revealed a heterogeneous set of miRNAs investigated, indicating that research focuses on a wide range of miRNAs, with the same marker rarely being reported across multiple studies.

Several studies have identified promising urinary miRNAs for noninvasive diagnosis of RCC. Despite variability in results and differences in study designs, populations and methodologies, some key patterns have emerged in terms of the most frequently studied miRNAs and general trends in their expression.

Whereas the miRNAs were not relevantly expressed in RCC patients could be due to fluctuations of urinary levels or laboratory equipment not sensitive enough to lower concentrations. The lack of internationally validated tests for laboratory investigations constituted an obstacle to this research field.

The role of urinary miRNAs as potential biomarker for follow-up in patients after adjuvant therapy has not been addressed in the literature yet, but once reliable miRNAs will be assessed, it could lead to interesting discoveries regarding potential response index to chemotherapy for RCC patients.

Another potential role for urinary miRNAs as predictors of metastases in higher stages of RCC is under investigation: in 2023, one study found that miR-191-5p, miR-324-3p, and miR-186-5p exhibited a strong association with metastasis development in patients with pathological T3 (pT3) tumors [24]. In our selection, only one study evaluated the correlation of urinary level of miRNAs in metastatic patients: urinary miR-30a-5p^me^ levels had an 80% sensitivity, 71% specificity, and 73% accuracy in differentiating patients with metastases (both synchronous and metachronous) from those without metastatic disease [20]. 

The main strength of our review is that, to our knowledge, it represents the first analysis specifically focused on urinary markers and the diagnosis of ccRCC. Another key strength of our review is the inclusion of rigorous methodological tools, including risk of bias assessment and certainty of evidence evaluation, approaches absent in prior reviews [23,24].

Compared to the review conducted by Aveta et al. [23], that analyzed the urinary miRNAs involved in all urological cancers, we adopted stricter selection criteria concerning circulating or cell-free miRNAs, focusing exclusively on RCC.

When compared to the previous review, our findings align in identifying significant miRNAs in patients with ccRCC, such as miRNA-15a 25, as well as the family of let-7 miRNAs.

Tito et al. also confirmed the role of miRNAs miR-122, miR-1271, miR-15b, and miR-210-3p, presenting results that align with the findings of our review [24].

In light of the latest advancements in RCC diagnosis, our review provides an updated overview of additional miRNAs, including miRNA-210, miR-30a-5p, miRNA-1275, let-7d-5p, miR-205-5p, miR-34a-5p, and miR-365a-3p.

Despite progress, the studies conducted so far have several limitations that restrict their clinical application. First, variability in miRNA extraction and quantification methodologies may affect the reproducibility of results. Second, the small clinical sample sizes used and the lack of large-scale multicenter studies limit the validity of the results obtained. In addition, the biological heterogeneity of RCC and the presence of subtypes with different molecular characteristics further complicate data interpretation.

One of the critical issues that has emerged from the previous systematic reviews published to date on urinary miRNAs is the lack of a specific selection of studies based on the histotype of RCC. This limitation has several important implications.

Indeed, one of the selection criteria of our review was to specifically investigate the diagnostic role of miRNAs in ccRCC. Although this is a methodological limitation in that it narrows the focus of the review, the absence of a specific selection of urinary miRNAs by histotype in previous reviews has led to heterogeneous results that do not accurately reflect biological differences between subtypes [25]. On the other hand, our approach is a strength because it allows for more accurate and targeted results and lays the foundation for validating more specific and reliable biomarkers for diagnosis and follow-up of patients with ccRCC, thus improving the clinical accuracy of these biomarkers.

The main limitations of this review included the limited number of databases considered for study selection and to evaluate only the miRNAs involved in RCC studies, excluding the potential association with other histological types.

Therefore, going forward, further large-scale clinical trials are needed to establish the role of urinary miRNAs as valid and reliable biomarkers. In order to improve future studies on the diagnostic role of miRNAs, the following suggestions could be helpful:Collaboration between research centers and clinical institutions is considered essential to validate these biomarkers in different populations and clinical settings.Studies evaluating urinary miRNAs multiple times in homogeneous populations could resolve the issue of intraindividual and interindividual fluctuation of miRNA in urine.Standardized procedures for data reporting and sample collecting must be created to enhance study comparability, increase the reliability of the results and avoid future inconsistencies among studies.Technological advances in miRNAs detection, such as next-generation sequencing and machine learning-based analysis, may increase sensitivity and specificity, improving the viability of miRNA-based diagnostics in standard clinical settings.The accuracy of diagnosis may be increased by combining miRNAs with additional biomarkers, such as protein or genetic biomarkers.Further studies with longer follow-up of pT3 and pT4 RCC patients would enrich our current knowledge about urinary miRNAs as potential predictors of metastases and response index to adjuvant therapy.To reduce risk of bias and increase the quality of future studies, it is crucial to identify and account for relevant confounding factors and apply the analysis to larger, independent cohorts to ensure their generalizability and clinical applicability.

## 5. Conclusions

Considering the variability and heterogeneity of the obtained results, as well as the vastness of the topic, expanding research in this field appears highly promising. To support further advancements, it would be useful to establish a database that consolidates international findings.

## Figures and Tables

**Figure 1 cancers-17-01336-f001:**
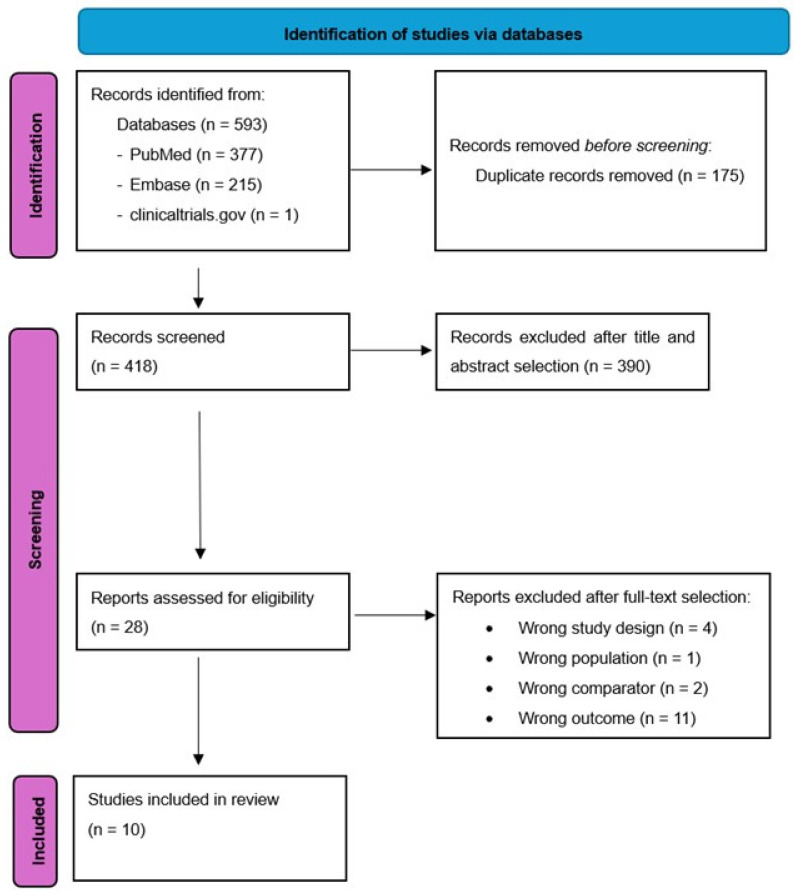
PRISMA 2020 flow diagram.

**Figure 2 cancers-17-01336-f002:**
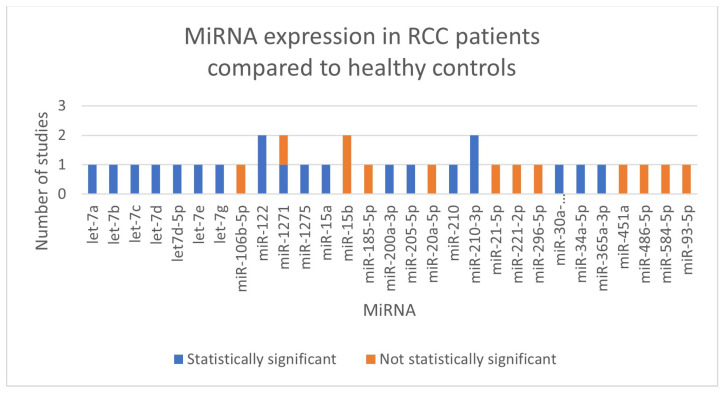
Summary of miRNAs expressed in our selection of studies.

**Table 1 cancers-17-01336-t001:** Included studies, main characteristics and summarized results. **RCC** (Renal cell carcinoma); **ccRCC** (clear cell RCC); **pRCC** (papillary RCC); **chRCC** (chromophobe RCC); **↑** (overexpressed); **↓** (underexpressed).

Study (First Author, Publication Year, Countries)	Number of Participants (Cases; Controls)	miRNA	Results
Bustos (2024, USA) [13]	34 (18 RCC; 16 healthy)	349 miRNAs evaluated; 9 miRNAs (miR-296-5p, miR-486-5p, miR-185-5p, miR-106b-5p, miR-451a, miR-93-5p, miR-584-5p, miR-20a-5p, and miR-1275) went through validation	miR-1275 (*p* = 3.71 × 10^−6^) ↑ in RCC (ccRCC, pRCC and cRCC) compared to controls
Cochetti (2020, Italy) [14]	27 (13 ccRCC; 14 healthy)	55 miRNAs evaluated; 3 miRNAs (miR-122, miR-15b, miR-1271) went through validation	miR-122 (*p* = 0.0042) and miR-1271 (*p* = 0.0101) ↑ in ccRCC compared to controls; miR-15b (*p* = 0.4094) no statistical difference between cases and controls
Cochetti (2022, Italy) [15]	56 (28 ccRCC; 28 healthy)	miR-122, miR-1271-5p, miR-15b-5p	miR-122 (*p* = 0.0192) ↑ in ccRCC compared to controls; miR-1271-5p (*p* = 0.0645) e miR-15b-5p (*p* = 0.0817) no statistical difference between cases and controls; the 7p-urinary score (parameter #1, miR-1271-5p; #2, miR-122-5p/miR-16-5p; #3, miR-122-5p/miRTC; #4, miR-1271-5p/miR-16-5p; #5, miR-1271-5p/miRTC; #6, miR-15b-5p/miRTC; #7, miR-15b-5p/Cel-miR-39-3p) showed statistical difference (*p* < 0.0001) between ccRCC and controls
Fedorko (2017, Czech Republic) [16]	105 (69 ccRCC; 36 healthy)	let-7 miRNAs (let-7a, let-7b, let-7c, let-7d, let-7e and let-7g)	let-7a (*p* < 0.001), let-7b (*p* < 0.001), let-7c (*p* = 0.005), let-7d (*p* = 0.006), let-7e (*p* = 0.015), and let-7g (*p* = 0.002) ↑ in ccRCC compared to controls
Li (2017, France, China) [17]	120 (75 ccRCC; 45 healthy)	miR-210	miR-210 (*p* < 0.001) ↑ in ccRCC compared to controls; miR-210 (*p* < 0.0001) ↓ in patients 1 week after nephrectomy compared to ccRCC pre-surgery
Mytsyk (2018, Ukraine) [18]	82 (22 ccRCC, 16 pRCC, 14 chRCC, 8 oncocytoma, 2 papillary adenoma, 5 angiomyolipoma; 15 healthy)	miR-15a	miR-15a ↑ in RCC compared to controls and benign tumors (*p* < 0.01); no significant difference (*p* > 0.05) in miR-15a expression levels between ccRCC, pRCC and chRCC subtypes; miR-15a (*p* < 0.01) ↓ in patients 8 days after nephrectomy compared to ccRCC pre-surgery
Oto (2021, Spain) [19]	115 (45 ccRCC, 16 pRCC, 6 chRCC; 13angiomyolipomas; 48 healthy)	179 miRNAs evaluated; 5 miRNAs (miR-200a-3p, let-7d-5p, miR-205-5p, miR-34a-5p and miR-36) went through validation	miR-200a-3p (*p* = 0.024), let-7d-5p (*p* = 0.035), miR-205-5p (*p* = 0.029), miR-34a-5p (*p* = 0.038) and miR-365a-3p (*p* = 0.001) ↑ in RCC compared to controls; no statistical difference (*p* > 0.05) between healthy controls and angiomyolipoma; let-7d-5p (*p* = 0.046), miR-152-3p (*p* = 0.023), miR-30c-5p (*p* = 0.042), miR-362-3p ( *p* = 0.03) and miR-30e-3p (*p* = 0.048) ↓ in patients 14 weeks after nephrectomy compared to RCC pre-surgery
Outeiro-Pinho (2020, Portugal) [20]	366 (224 ccRCC; 142 healthy)	miR-30a-5p^me^	miR-30a-5p^me^ (*p* <0.0001) ↑ in ccRCC compared to controls
Petrozza (2017, Italy) [21]	48 (38 ccRCC; 10 healthy)	miR-21-5p, miR-210-3p and miR-221-3p	miR-210-3p (*p* < 0.01) ↑ in ccRCC compared to controls; miR-21-5p and miR-221-3p no statistical difference (*p* > 0.05) between cases and controls; miR-210-3p (*p* < 0.05) ↓ in patients 6 months after nephrectomy compared to ccRCC pre-surgery
Petrozza (2020, Italy) [22]	37 (21 ccRCC; 16 healthy)	miR-210-3p	miR-210-3p (*p* < 0.05) ↑ in ccRCC compared to controls; follow up: miR-210-3p (*p* < 0.05) ↓ in ccRCC after nephrectomy compared to ccRCC pre-surgery

**Table 2 cancers-17-01336-t002:** Risk of bias and certainty assessment of included studies. ^1^—no confounding factors mentioned; ^2^—small cohort (<200 participants between patients and controls considering all phases of the study).

	Bustos (2024) [13]	Cochetti (2020) [14]	Cochetti (2022) [15]	Fedorko (2017) [16]	Li (2017) [17]	Mytsyk (2018) [18]	Oto (2021) [19]	Outeiro-Pinho (2020) [20]	Petrozza (2017) [21]	Petrozza (2020) [22]
**Risk of bias (ROBINS-E)**										
Domain 1 (Risk of bias due to confounding)	Some concerns ^1^	Low risk	Low risk	Low risk	Some concerns ^1^	Some concerns ^1^	Low risk	Low risk	Some concerns ^1^	Some concerns ^1^
Domain 2 (Risk of bias arising from measurement of the exposure)	Low risk	Low risk	Low risk	Low risk	Low risk	Low risk	Low risk	Low risk	Low risk	Low risk
Domain 3 (Risk of bias in selection of participants into the study)	Low risk	Low risk	Low risk	Low risk	Low risk	Low risk	Low risk	Low risk	Low risk	Low risk
Domain 4 (Risk of bias due to post-exposure interventions)	Low risk	Low risk	Low risk	Low risk	Low risk	Low risk	Low risk	Low risk	Low risk	Low risk
Domain 5 (Risk of bias due to missing data)	Low risk	Low risk	Low risk	Low risk	Low risk	Low risk	Low risk	Low risk	Low risk	Low risk
Domain 6 (Risk of bias arising from measurement of the outcome)	Low risk	Low risk	Low risk	Low risk	Low risk	Low risk	Low risk	Low risk	Low risk	Low risk
Domain 7 (Risk of bias in selection of the reported result)	Low risk	Low risk	Low risk	Low risk	Low risk	Low risk	Low risk	Low risk	Low risk	Low risk
Overall risk of bias	Some concerns	Low risk	Low risk	Low risk	Some concerns	Some concerns	Low risk	Low risk	Some concerns	Some concerns
**Certainty assessment (GRADE)**										
Inconsistency	Not serious	Not serious	Not serious	Not serious	Not serious	Not serious	Not serious	Not serious	Not serious	Not serious
Indirectness	Not serious	Not serious	Not serious	Not serious	Not serious	Not serious	Not serious	Not serious	Not serious	Not serious
Imprecision	Serious ^2^	Serious ^2^	Serious ^2^	Serious ^2^	Serious ^2^	Serious ^2^	Serious ^2^	Not serious	Serious ^2^	Serious ^2^
Other considerations	None	None	None	None	None	None	None	None	None	None
**Quality**	**Low**	**Moderate**	**Moderate**	**Moderate**	**Low**	**Low**	**Moderate**	**High**	**Low**	**Low**

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
