# Peer review of "Evaluation of Urinary miRNA in Renal Cell Carcinoma: A Systematic Review"

_cancers, 2025, doi:10.3390/cancers17081336_

Round 1

Reviewer 1 Report

Comments and Suggestions for Authors

Comments:

  • Is there a correlation between the same miRNA in serum and urine?
  • Can the chemical-physical caracteristics of urine alter the stability of miRNAs?
  • Absence of validated test to determine and compare miRNAs.
  • Biological heterogeneity of RCC has been described in the same patient and between patients, how to adress this problem?
  • Can these findings be useful in the follow up of patients after adjuvant therapy in RCC?
  • Ca urine-miRNA be potential markers to predict the occurrence of metastases in pT3 RCC?
  • Is miRNA biomarker panel reproducible across centers taking bias into account?
  • Do you have direct experience with urine-miRNA?

Author Response

Comments 1: Is there a correlation between the same miRNA in serum and urine?

Response 1: Thank you for the question. Yes, as we state in the discussion, both serum and urinary miR-210 are upregulated in RCC compared to controls; we also added a reference to our recent systematic review about serum miRNAs expression in RCC.

Comments 2: Can the chemical-physical caracteristics of urine alter the stability of miRNAs?

Response 2: Thank you for your comment. The chemical-physical characteristics of urine such as pH, ionic strength, and the presence of RNases can alter the stability of miRNAs, but there are some effective miRNAs’ protective mechanisms. The most important are encapsulation of miRNAs in exosomes or their association with RNA-binding proteins (Argonaute proteins), that prevent their degradation. So, we state that in our manuscript: “Importantly, circulating miRNAs have been shown to be remarkably stable in bodily fluids, such as blood and urine, making them ideal candidates for non-invasive diagnostic testing.”

Comment 3: Absence of validated test to determine and compare miRNAs.

Response 3: Thank you for your comment. We added this sentence to the Discussion section: “Whereas the miRNAs were not relevantly expressed in RCC patients could be due to fluctuations of urinary levels or laboratory equipment not sensitive enough to lower concentrations. The lack of internationally validated tests for laboratory investigations constituted an obstacle to this research field.”

Comment 4: Biological heterogeneity of RCC has been described in the same patient and between patients, how to adress this problem?

Response 4: Thank you for letting us focus more on this. We added a bullet point list to the Discussion section where we propose different solutions to uniform and increase the quality of future studies. We addressed this problem in the following sentences: “Studies evaluating urinary miRNAs multiple times in homogeneous populations could resolve the issue of intraindividual and interindividual fluctuation of miRNA in urine. Standardized procedures for data reporting and sample collecting must be created to enhance study comparability, increase the reliability of the results and avoid future inconsistencies among studies. Technological advances in miRNAs detection, such as next-generation sequencing and machine learning-based analysis, may increase sensitivity and specificity, improving the viability of miRNA-based diagnostics in standard clinical settings.

Comments 5: Can these findings be useful in the follow up of patients after adjuvant therapy in RCC?

Response 5: Thank you for the suggestion. We didn’t find any study evaluating urinary miRNAs in our selection, but a potential role in follow-up of RCC patients after adjuvant therapy could be an interesting objective for a future study. We added in the discussion section this sentence addressing the issue: “The role of urinary miRNAs as potential biomarker for follow-up in patients after adjuvant therapy has not been addressed in literature yet, but once reliable miRNAs will be assessed, it could lead to interesting discoveries regarding potential response index to chemotherapy for RCC patients.”

Comments 6: Ca urine-miRNA be potential markers to predict the occurrence of metastases in pT3 RCC?

Response 6: Thank you for your comment. A study focused on predicting pT3 RCC metastases was found in our selection of studies but ultimately excluded for lack of adherence to our inclusion criteria because there were no healthy controls (it can be found in the list of excluded studies in the supplementary materials). We added a reference to it in the Discussion section and underlined the only study in our final selection addressing metastatic patients: “Another potential role for urinary miRNAs as predictors of metastases in higher stages of RCC is under investigation: in 2023, one study found that miR-191-5p, miR-324-3p, and miR-186-5p exhibited a strong association with metastasis development in patients with pathological T3 (pT3) tumors [24]. In our selection, only one study evaluated the correlation of urinary level of miRNAs in metastatic patients: urinary miR-30a-5pme levels had an 80% sensitivity, 71% specificity, and 73% accuracy in differentiating patients with metastases (both synchronous and metachronous) from those without metastatic disease [20].

Comments 7: Is miRNA biomarker panel reproducible across centers taking bias into account?

Response 7: Thanks for your valuable question. MiRNA biomarker panel may have the potential to achieve a high reproducibility rate, but it requires strict standardized protocols and multicenter validation studies. The most controversial point is the use of a standardized normalization method, which can be solved with the introduction of a robust and validated endogenous miRNA or the use of specific synthetic spike-in controls, in order to improve reproducibility across different centers. Aiming to steer future studies toward this research, we state in our manuscript: “First, variability in miRNA extraction and quantification methodologies may affect the reproducibility of results. Second, the small clinical sample sizes used and the lack of large-scale multicenter studies limit the validity of the results obtained. In addition, the biological heterogeneity of RCC and the presence of subtypes with different molecular characteristics further complicate data interpretation”.

Comments 8: Do you have direct experience with urine-miRNA?

Response 8: Thanks for your question. Yes, we do have direct laboratory experience with urine-miRNA. We have measured their levels in urine of patients with renal cancer comparing them with healthy controls, and we have published two research articles on this subject:

  1. Cochetti G, Cari L, Nocentini G, Maulà V, Suvieri C, Cagnani R, Rossi De Vermandois JA, Mearini E. Detection of urinary miRNAs for diagnosis of clear cell renal cell carcinoma. Sci Rep. 2020 Dec 4;10(1):21290. doi: 10.1038/s41598-020-77774-9. PMID: 33277569; PMCID: PMC7718885.
  2. Cochetti G, Cari L, Maulà V, Cagnani R, Paladini A, Del Zingaro M, Nocentini G, Mearini E. Validation in an Independent Cohort of MiR-122, MiR-1271, and MiR-15b as Urinary Biomarkers for the Potential Early Diagnosis of Clear Cell Renal Cell Carcinoma. Cancers (Basel). 2022 Feb 22;14(5):1112. doi: 10.3390/cancers14051112. PMID: 35267420; PMCID: PMC8909007.

Reviewer 2 Report

Comments and Suggestions for Authors

This manuscript provides a well-structured and insightful analysis of an important and evolving area of oncological research.

The study effectively highlights the complexity and inconsistencies in existing findings while emphasizing the need for standardized protocols to unify global research efforts. The identification of key miRNAs, particularly miR-210 and the let-7 family, as promising biomarkers is valuable, as is the discussion of post-nephrectomy expression changes. Additionally, your emphasis on the heterogeneity of current studies underscores the challenges in conducting a meta-analysis, making a strong case for the creation of a consolidated international database.

To further strengthen the manuscript, authors may consider expanding on potential strategies for standardizing normalization methods and addressing histological subtype variations. 

Author Response

Comments 1: This manuscript provides a well-structured and insightful analysis of an important and evolving area of oncological research.

The study effectively highlights the complexity and inconsistencies in existing findings while emphasizing the need for standardized protocols to unify global research efforts. The identification of key miRNAs, particularly miR-210 and the let-7 family, as promising biomarkers is valuable, as is the discussion of post-nephrectomy expression changes. Additionally, your emphasis on the heterogeneity of current studies underscores the challenges in conducting a meta-analysis, making a strong case for the creation of a consolidated international database.

Response 1: Thank you for your valuable words.

Comments 2: To further strengthen the manuscript, authors may consider expanding on potential strategies for standardizing normalization methods and addressing histological subtype variations.

Response 2: Thank you for your suggestions, we added a bullet point list at the end of the Discussion section to help standardize future research in this field: “In order to improve future studies on the diagnostic role of miRNAs, the following suggestions could be helpful:

  • Collaboration between research centers and clinical institutions is considered essential to validate these biomarkers in different populations and clinical settings.
  • Studies evaluating urinary miRNAs multiple times in homogeneous populations could resolve the issue of intraindividual and interindividual fluctuation of miRNA in urine.
  • Standardized procedures for data reporting and sample collecting must be created to enhance study comparability, increase the reliability of the results and avoid future inconsistencies among studies.
  • Technological advances in miRNAs detection, such as next-generation sequencing and machine learning-based analysis, may increase sensitivity and specificity, improving the viability of miRNA-based diagnostics in standard clinical settings.
  • The accuracy of diagnosis may be increased by combining miRNAs with additional biomarkers, such as protein or genetic biomarkers.
  • Further studies with longer follow-up of pT3 and pT4 RCC patients would enrich our current knowledge about urinary miRNAs as potential predictors of metastases and response index to adjuvant therapy.
  • To reduce risk of bias and increase the quality of future studies, it is crucial to identify and account for relevant confounding factors and apply the analysis to larger, independent cohorts to ensure their generalizability and clinical applicability.”

Reviewer 3 Report

Comments and Suggestions for Authors

i) The authors aimed at performing a meta-analysis of the main urinary miRNAs implicated in RCC but decided not to perform it because of the expected excessive heterogeneity between studies. The question to be raised is why did the authors advance in the study if they clearly identified such problem?
ii) The authors point out some main limitations associated with the study performed. One of such limitations they refer being the limited number of databases considered for study selection. I cannot understand this limitation. If the authors recognize that such was a limitation why they did not expand the search to other databases.
iii) The authors decided to perform a systematic analysis instead of a meta-analysis based on the limitations of the study to advance for a meta-analysis. In my opinion the analysis performed is rather minimalist and cannot anticipate any major impact to the field of research. 

Author Response

Comments 1: The authors aimed at performing a meta-analysis of the main urinary miRNAs implicated in RCC but decided not to perform it because of the expected excessive heterogeneity between studies. The question to be raised is why did the authors advance in the study if they clearly identified such problem?

Response 1: Thank you for let us be more clear on the findings of our work. We added a figure (Figure 2) to further underscore the heterogeneity between studies. Unfortunately, only a couple of miRNAs were investigated in more than one study and those studies were considering overlapping populations (same authors and setting, only larger cohort).

Comments 2: The authors point out some main limitations associated with the study performed. One of such limitations they refer being the limited number of databases considered for study selection. I cannot understand this limitation. If the authors recognize that such was a limitation why they did not expand the search to other databases.

Response 2: Thank you for the comment. Unfortunately, we did not have access to other databases, such as Cochrane Library, so our research was limited to the databases listed in the manuscript. Nevertheless, we have performed a deep search in the literature on most of the main databases available.

Comments 3: The authors decided to perform a systematic analysis instead of a meta-analysis based on the limitations of the study to advance for a meta-analysis. In my opinion the analysis performed is rather minimalist and cannot anticipate any major impact to the field of research.

Response 3: Thank you for your valuable comment. The authors believe that this review could be considered a starting point to carry on future research in this field.
We also added a bullet point list of suggestion to help increase the number and quality of future works: “In order to improve future studies on the diagnostic role of miRNAs, the following suggestions could be helpful:

  • Collaboration between research centers and clinical institutions is considered essential to validate these biomarkers in different populations and clinical settings.
  • Studies evaluating urinary miRNAs multiple times in homogeneous populations could resolve the issue of intraindividual and interindividual fluctuation of miRNA in urine.
  • Standardized procedures for data reporting and sample collecting must be created to enhance study comparability, increase the reliability of the results and avoid future inconsistencies among studies.
  • Technological advances in miRNAs detection, such as next-generation sequencing and machine learning-based analysis, may increase sensitivity and specificity, improving the viability of miRNA-based diagnostics in standard clinical settings.
  • The accuracy of diagnosis may be increased by combining miRNAs with additional biomarkers, such as protein or genetic biomarkers.
  • Further studies with longer follow-up of pT3 and pT4 RCC patients would enrich our current knowledge about urinary miRNAs as potential predictors of metastases and response index to adjuvant therapy.
  • To reduce risk of bias and increase the quality of future studies, it is crucial to identify and account for relevant confounding factors and apply the analysis to larger, independent cohorts to ensure their generalizability and clinical applicability.”

Round 2

Reviewer 3 Report

Comments and Suggestions for Authors

The revisions made by the authors did not improve the maniscript from its originla version. The addition of figure 2 and the suggestions made do not justify consideration for publication. 

Author Response

Comment 1: The revisions made by the authors did not improve the maniscript from its originla version. The addition of figure 2 and the suggestions made do not justify consideration for publication. 

Response 1: Dear Reviewer, 
We sincerely thank you for your valuable comments and for the opportunity to clarify the scope and methodology of our work.
We would like to emphasize that this study is a systematic review, and, as such, it does not aim to introduce new findings, but rather to identify, appraise, and synthesize the current body of literature on urinary microRNAs. To date, this is the first systematic review specifically focused on this topic.
Due to the marked heterogeneity of the included studies and the variety of methodological approaches, it was unfortunately not feasible to conduct a meta-analysis. However, this heterogeneity itself provides relevant insights, as it reveals the lack of standardization in the field and underscores the need for more robust and harmonized study designs.
Our review thus serves a dual purpose: it summarizes the existing evidence and delineates the critical gaps that need to be addressed in future research to enable the clinical translation of urinary miRNAs, especially in the oncological setting, where their use is still at an early stage.
Thank you again for your thoughtful feedback.